# Mitochondria-Associated Endoplasmic Reticulum Membrane (MAM) Is a Promising Signature to Predict Prognosis and Therapies for Hepatocellular Carcinoma (HCC)

**DOI:** 10.3390/jcm12051830

**Published:** 2023-02-24

**Authors:** Yuyan Chen, Senzhe Xia, Lu Zhang, Xueqian Qin, Zhengyi Zhu, Tao Ma, Shushu Lu, Jing Chen, Xiaolei Shi, Haozhen Ren

**Affiliations:** 1Department of Hepatobiliary Surgery, Affiliated Drum Tower Hospital of Nanjing University Medical School, Nanjing 210008, China; 2Department of Hepatobiliary Surgery, Nanjing Drum Tower Hospital, Clinical College of Jiangsu University, Nanjing 210008, China; 3Department of Hepatobiliary Surgery, Nanjing Drum Tower Hospital, Clinical College of Xuzhou Medical University, Nanjing 210009, China; 4Department of Hepatobiliary Surgery, Nanjing Drum Tower Hospital, Clinical College of Traditional Chinese and Western Medicine, Nanjing University of Chinese Medicine, Nanjing 210009, China; 5Department of Gastroenterology, Affiliated Hospital of Nantong University, Nantong 226001, China; 6Department of Hepatobiliary Surgery, Nanjing Drum Tower Hospital, Clinical College of Nanjing Medical University, Nanjing 210009, China

**Keywords:** hepatocellular carcinoma, mitochondria-associated endoplasmic reticulum membrane, tumor microenvironment, immune therapy, chemotherapy

## Abstract

Background: The roles of mitochondria and the endoplasmic reticulum (ER) in the progression of hepatocellular carcinoma (HCC) are well established. However, a special domain that regulates the close contact between the ER and mitochondria, known as the mitochondria-associated endoplasmic reticulum membrane (MAM), has not yet been investigated in detail in HCC. Methods: The TCGA-LIHC dataset was only used as a training set. In addition, the ICGC and several GEO datasets were used for validation. Consensus clustering was applied to test the prognostic value of the MAM-associated genes. Then, the MAM score was constructed using the lasso algorithm. In addition, uncertainty of clustering in single-cell RNA-seq data using a gene co-expression network (AUCell) was used for the detection of the MAM scores in various cell types. Then, CellChat analysis was applied for comparing the interaction strength between the different MAM score groups. Further, the tumor microenvironment score (TME score) was calculated to compare the prognostic values, the correlation with the other HCC subtypes, tumor immune infiltration landscape, genomic mutations, and copy number variations (CNV) of different subgroups. Finally, the response to immune therapy and sensitivity to chemotherapy were also determined. Results: First, it was observed that the MAM-associated genes could differentiate the survival rates of HCC. Then, the MAM score was constructed and validated using the TCGA and ICGC datasets, respectively. The AUCell analysis indicated that the MAM score was higher in the malignant cells. In addition, enrichment analysis demonstrated that malignant cells with a high MAM score were positively correlated with energy metabolism pathways. Furthermore, the CellChat analysis indicated that the interaction strength was reinforced between the high-MAM-score malignant cells and T cells. Finally, the TME score was constructed, which demonstrated that the HCC patients with high MAM scores/low TME scores tend to have a worse prognosis and high frequency of genomic mutations, while those with low MAM scores/high TME scores were more likely to have a better response to immune therapy. Conclusions: MAM score is a promising index for determining the need for chemotherapy, which reflects the energy metabolic pathways. A combination of the MAM score and TME score could be a better indicator to predict prognosis and response to immune therapy.

## 1. Introduction

Hepatocellular carcinoma (HCC) is the most common (70–90%) liver cancer and the fifth most common cancer globally [1]. Although the five-year survival rate of early radical surgery for HCC can reach 75%, no typical symptoms are generally observed in the early stages of HCC. Most of the patients are already in the advanced stage when they are diagnosed, sometimes accompanied by distant metastasis, resulting in poor survival rates [2]. In the past few years, there has been some progress in the clinical application of drugs, such as sorafenib and lenvatinib, for the treatment of liver cancer [3]. Although these drugs can prolong the prognosis of patients to some extent, multi-center studies have demonstrated that these drugs have little effect on the long-term survival of HCC patients because of side effects such as chemotherapy resistance [4]. Therefore, early diagnosis is crucial for the long-term survival of HCC patients. At present, the prediction of prognosis in HCC patients depends on the tumor size, the number of lymph nodes, and distant metastasis [5]. However, HCC patients showed different outcomes, even with the same clinical characteristics. Several studies have demonstrated that HCC is highly heterogeneous due to its complex regulatory mechanisms [6,7]. Therefore, it is beneficial to predict the long-term survival of patients by exploring the mechanisms underlying the occurrence and development of HCC.

In recent years, with the extensive development of its clinical trials, tumor immunotherapy has demonstrated primary benefits, and immunotherapy for advanced HCC has become a popular area of research [8]. With the development of single-cell sequencing technology, several studies have demonstrated the unique functions of different cell types in HCC. Consequently, several immune therapies targeted at cell types, such as CAR-T and DC-CIK, have shown their effectiveness in various tumors, especially in some blood cancers [9]. However, the effects of these therapies on most solid tumors fail to live up to the expectations because of the T cell exhaustion in the tumor microenvironment (TME) [10,11]. Hence, a deeper understanding of TME shaped by the tumor can be beneficial for the immunotherapy of HCC patients. As in a typical inflammatory type of tumor, immune tolerance and escape mechanisms play important roles in the HCC progression [12]. In TME, cancer cells, cytokines, immune cells, and the extracellular matrix constitute a dynamic and complex system. Thus, as the oncogenic and anti-tumor effects exist and interact at the same time, we considered that investigating the regulatory mechanisms within the TME may help predict the prognosis and immune responses of HCC patients.

The endoplasmic reticulum (ER) and mitochondria, the two typical dynamic organelles, play important roles in several cellular activities. The mitochondria-associated endoplasmic reticulum membrane (MAM) is a specialized domain that functions as membrane contact sites between the ER and mitochondria and was first discovered by J. Vance et al. in 1990 [13]. Further study with electron microscopes showed that the distance between the mitochondria outer membrane and endoplasmic reticulum remains at about 10–25 nm, but they do not fuse, maintaining the organelles’ unique structure, and provide a functional platform with various enrichment proteins. Notably, MAM mediates communication between the two organelles and is involved in the exchange of proteins and metabolites, including Ca^2+^ and reactive oxygen species (ROS) [14]. In addition, the distance between the ER and mitochondria is a key checkpoint regulating cellular functions. Several studies have observed that close contact between the ER and mitochondria led to mitochondrial Ca^2+^ overload and apoptosis. In other words, the destruction of ER–mitochondrial contact points could stimulate mitochondrial oxidative respiration and ATP production [15,16]. Furthermore, MAM could also play an important role in tumor progression. Wang et al. observed that FUNDC1-dependent MAM is involved in angiogenesis in cancers [17]. Li et al. observed that GRP75-facilitated MAM controls cisplatin resistance, thus shortening the prognosis in ovarian cancer patients [18]. However, the effects of MAM on HCC progression and the underlying mechanism still remain unclear.

In this study, due to the strong heterogeneity of HCC, targeted therapy has not been very effective. In this study, to understand the potential mechanisms regulating the progression of HCC, we generated a MAM score to predict the prognosis and chemotherapy response of patients. Finally, we investigated the potential mechanisms underlying the impact of MAM structure on HCC progression by using single-cell data. Therefore, the current study provides a novel basis for the treatment of HCC.

## 2. Materials and Methods

### 2.1. Public Data Collection Acquisition

The expression of the MAM-related genes in the HCC and normal groups was analyzed by downloading several GEO microarray datasets, including GSE14520, GSE25097, GSE36376, GSE39791, GSE45436, GSE54236, GSE76427, GSE77314, GSE102083, and GSE 112790 from the GEO database “https://www.ncbi.nlm.nih.gov/geo/ (accessed on 12 January 2023)”. In addition, the clinical data of GSE14520 were also analyzed. In particular, the MAM scores in the transcatheter arterial chemoembolization (TACE) response and no response groups were analyzed based on GSE104580. The differences in the MAM scores of nine HCC single-cell samples were obtained from GSE125449. The detailed quality control procedure of individual cells is described in the “Single-cell data analysis” section below. The transcriptional data and clinical data were downloaded from the International Cancer Genome Consortium (ICGC) LIRI-JP database “https://dcc.icgc.org/ (accessed on 12 January 2023)”. The transcriptional data, clinical data, copy number variation (CNV) data, and mutation data were downloaded from the TCGA-LIHC database using the R package “TCGAbiolinks”. Additionally, the data from the Cancer Cell Line Encyclopedia (CCLE) were downloaded from the CCLE website “https://sites.broadinstitute.org/ccle (accessed on 12 January 2023)” for testing the drug sensitivity of the low- and high-MAM-score groups. 

### 2.2. Consensus Clustering Analysis

The patients were divided into two clusters in the TCGA, ICGC, and GSE14520 datasets using the R package “ConsensusClusterPlus” for validating whether the MAM-related genes could influence the prognosis of HCC patients.

### 2.3. Comparison of the Different Molecular Classifications of HCC

The detailed methods have been described by Shu et al. [19]. Briefly, the generated classifications in this study were compared with other classifications based on the TCGA-LIHC database in previous studies. 

### 2.4. MAM-Score and TME-Score Calculations

The MAM scores were calculated for five genes based on lasso Cox regression using the R package “glmnet”. A MAM score was calculated as follows: (−0.002405788) × ACAT1 + (0.005653644) × PACS2 + (0.001473221) × VDAC1 + (0.028582795) × MFN1 + (0.011694457) × ATAD3A. The HCC samples were divided into low and high groups based on the median MAM score in the TCGA HCC dataset and ICGC-LIRI-JP cohort. The CIBERSORT algorithm was initially used to assess the 22 immune cell type scores of each sample of the TCGA and ICGC datasets [20], and then a TME score was calculated as follows: (−5.860781) × activated NK cells + (−3.393689) × resting memory CD4+ T cells + (−3.283541) × CD8+ T cells. In addition, when a sample was identified as belonging to the “high-MAM-score/high-TME-score” or “low-MAM-score/low-TME-score” subgroups, it was recognized as a “mixed” group.

### 2.5. Prognostic Model Validation

The Kaplan–Meier (KM) analysis, time-dependent receiver operating characteristic (ROC) curves, univariate and multivariate Cox regression analyses, and nomogram were analyzed using the R packages “survival”, “survminer”, and “survivalROC” for evaluating the HCC prognosis prediction [21]. 

### 2.6. Single-Cell Data Analysis

The cell control quality criteria were set as follows: cell counts > 200, percent of mitochondria < 10, and percent of erythrocytes < 3. A total of 3821 cells from nine samples were finally identified by R package “Seurat”. “LogNormalize” function in the R package Seurat was used for standardizing the expression at the single-cell level. Further, the principal component analysis (PCA) and the subsequent t-distributed stochastic neighbor embedding (T-SNE) analysis were performed for the dimension reduction of the single-cell data. In addition, the cell types provided by the authors of this GEO series were used. As a result, the cells were divided into 7 cell types: cancer-associated fibroblasts (CAF), tumor-associated endothelial cells (TEC), tumor-associated macrophages (TAM), hepatic progenitor cells (HPC-like), malignant cells, T cells, and B cells. 

### 2.7. Single-Cell Gene Set Enrichment Analysis

The Uncertainty of Clustering in single-cell RNA-seq data using a gene co-expression network (AUCell) analysis was used to calculate the MAM scores of the single-cell gene sets [22]. For enrichment analysis, the marker genes of all the cell types were initially calculated, and then ToppGene “https://toppgene.cchmc.org/enrichment.jsp (accessed on 12 January 2023)” was used for the analysis of the enriched pathways of each cell type. In addition, the R package “scMetabolism” was used to predict the metabolism score for the cell types [23].

### 2.8. Mapping Specific Regulon Networks by SCENIC 

The R package “SCENIC” was applied to analyze the activated regulons of each cell type following the standard procedure [22]. Then, “runGenie3” and “RcisTarget” analyses were performed for determining the potential transcription factors of each cell type. In addition, the regulon activity score was calculated by R package “AUCell” and represented by “T-SNE” plots. 

### 2.9. CellChat Analysis

R package “CellChat” was applied to determine the co-communications between the malignant cells and other cell types in the low- and high-MAM-score groups. In addition, the differences in the receptor ligands in the low- and high MAM-malignant cells were visualized using the “netVisual-bubble” function.

### 2.10. Regulon Analysis

As previously described, the R package “RTN” was used for constructing the transcriptional regulatory regulons of 17 transcriptional factors (TFs), which regulated the malignant cells, CAF, and TEC, in the low- and high MAM-groups [24]. 

### 2.11. Calculation of Microenvironment Cell Abundance and Immune Infiltration in the Low- and High-MAM-Score Groups

First, CIBERSORT and ESTIMATE algorithms were used for calculating the MAM and TME scores related to specific microenvironment cells, and comparing the immune checkpoints, immune cell proportions, and several immune processes. In addition, the signature scores of the MAM–TME subgroups were calculated as previously described to evaluate the differences in the distribution of microenvironment-related signatures of different groups [25].

### 2.12. Prediction of the Response to Immunotherapy in HCC

The exclusion scores and the responses to TIDE therapy are available at “http://tide.dfci.harvard.edu/ (accessed on 12 January 2023)”. Then, the TIDE scores were used to calculate and determine whether each sample from the different groups is sensitive to PD1 and CTAL4 treatment in the submap platform, which is an algorithm for drawing inferences by comparing the similarities between the expression profiles “https://cloud.genepattern.org/gp (accessed on 12 January 2023)”. In addition, the Cancer Immunome Atlas (TCIA) database was used for obtaining the immunephenoscore (IPS) data from HCC patients. Subsequently, the IPS differences between the high-MAM/low-TME-scores, low-MAM/high-TME-scores, and the mixed subgroups were analyzed using the boxplots.

### 2.13. Prediction of the Response to Chemotherapy in HCC

To determine the drug sensitivity of the cancer cell line, the PRISM dataset “https://depmap.org/portal/prism/ (accessed on 12 January 2023)” and Cancer Therapeutics Response Portal (CTRP, “https://portals.broadinstitute.org/ctrp (accessed on 12 January 2023)”) were used for calculating the area under the curve (AUC) and the correlations between the low- and high-MAM-score groups. In addition, the CMAP platform “clue.io/query (accessed on 12 January 2023)” was also used for evaluating the drug susceptibility differences between the low-MAM-score and high-MAM-score subgroups.

### 2.14. Statistical Analysis

All the statistical data analyses were conducted using R software 4.1.3, and most of the plots were obtained using the R package “ggplot2” [26]. Some of the plots were drawn using the Hiplot “https://hiplot-academic.com (accessed on 12 January 2023)”. The data were evaluated using Wilcoxon tests when two groups were compared, and Kruskal–Wallis tests were used when three groups were compared. Statistical significance was described as follows: ns, not significant; * *p* < 0.05; ** *p* ≤ 0.01; *** *p* ≤ 0.001; and **** *p* ≤ 0.0001.

### 2.15. Work Flow

In this study, we investigated the expression of the MAM-related genes in several public datasets. Further, the MAM score was generated and validated in the TCGA and ICGC datasets. Subsequently, the single-cell data were used and it was observed that the MAM score was higher in the malignant cells and was closely related to energy metabolic pathways. In addition, the study results indicated that patients with high MAM scores tend to show a better response to chemotherapy. Furthermore, the TME scoring system (TME score) was constructed, which further helps the MAM score to classify the tumor immune infiltration environment. Finally, the results indicated that the HCC patients with a high MAM score/low TME score tend to have a worse prognosis and high frequency of genomic mutations, while patients with a low MAM score/high TME score were more likely to respond better to immune therapy. The workflow is illustrated in Figure 1.

## 3. Results

### 3.1. Identification of the Expression and Prognostic Value of MAM in HCC

MAM functions as a dynamic organelle for the transmission of ATP, Ca^2+^, and ROS, and is illustrated in Figure 2A. The commonly identified MAM resident proteins are involved in a series of cytological functions. In this study, several HCC datasets were identified by analyzing the expression of these genes between the HCC and normal groups. However, some genes could not be detected due to the limitations of microarray. It was observed that the expression of several genes in HCC patients was different from that in the normal samples (Figure 2B). Moreover, for investigating the clinical indices of the MAM molecular subtypes in HCC, the samples were divided into two groups (C1 and C2) according to the optimal cluster number (k = 2) in the TCGA, ICGC, and GSE14520 datasets. Notably, it was observed that MAM subtypes in TCGA were closely related to other HCC subtypes, including *LEE*, *Hoshida*, *Boyault*, and *Chiang* (Figure 2C). In addition, the KM analysis was used for investigating the prognostic value of the C2 and C1 groups and it was observed that the MAM subtypes could reasonably have the potential to predict the survival rates of HCC patients in the TCGA, ICGC, and GSE14520 datasets (Figure 2D–F). Together, these results revealed the MAM-related genes could distinguish the HCC samples and predict the poor prognosis of HCC patients.

### 3.2. Construction and Verification of the MAM-Based Score in HCC

The lasso algorithm was applied for the construction of the MAM score model in TCGA, which was used to predict the prognostic value of the 28 MAM-related genes in HCC samples. Finally, five genes, ACAT1, PACS2, VDAC1, MFN1, and ATAD3A, were identified. In this study, the TCGA dataset was used as a training set, and the ICGC dataset was used as a validation set. First, the distribution of the expression of five MAM-related genes, survival status, MAM scores, and KM curve between the training and verification sets were plotted (Figure 3A,B). In addition, the correlations between the expressions of five MAM-related genes and MAM scores were examined in this study, and the results indicated that while ACAT1 was negatively correlated with the other four genes, the other four genes were positively correlated with each other (Figure 3C). Then, the prognostic value of the MAM score was determined through the KM analysis, and the results indicated that the high-MAM-score subgroup of HCC patients had shorter overall survival (OS) than the low-MAM-score subgroup (TCGA: *p* = 0.0004; ICGC: *p* = 0.0004, Figure 3D,E). Further, a time-dependent ROC curve was used for evaluating the predictive ability of the MAM score. The AUC in the training set was 0.61 at one year, 0.64 at three years, 0.67 at four years, and 0.75 at five years, while the AUC in the validation set was 0.70 at one year, 0.70 at three years, 0.72 at four years, and 0.73 at five years (Figure 3F,G). Furthermore, the univariate and multivariate analysis revealed that high MAM score was an independent factor for short OS in the training set (*p* = 0.0019; HR = 1.8; 95% CI: 1.2–2.6, Figure 3H–J) and the validation set (*p* = 0.002; HR = 2.9; 95% CI: 1.5–5.7, Figure 3I–K). Lastly, a nomogram was generated to examine the MAM scores and other clinical characteristics, indicating higher clinical use of the nomogram relative to the MAM score and TNM stage in OS prediction (Figure 3L,M). Thus, these results suggested that the MAM score might serve as a candidate prognostic factor in HCC.

### 3.3. Identification of the MAM Score and Its Potential Enriched Pathways at the Single-Cell Level 

Firstly, the cells were classified into seven types, including CAF, TEC, TAM, HPC-like, malignant, T, and B cells, based on a previous study [27]. As shown in Figure 4A, these cells were visualized by T-SNE. To further evaluate the MAM score at the single-cell level, the AUCell was used to analyze the MAM scores in the seven cell types, and it was observed that the MAM scores in the non-immune cell types (CAF, TEC, HPC-like, and malignant cells), especially the malignant cells, were higher than those in the immune cell types (TAM, T, and B cells; Figure 4B,C). The enrichment analysis conducted to investigate the unique functions of each cell type revealed that TEC is involved in several pathways linked with VEGFR, CAF plays a critical role in the extracellular organization, TAM plays a key role in the innate immune system, T and B cells tend to influence several immune-related pathways, and malignant cells are closely correlated with several metabolic pathways, such as ATP synthesis and the electron transport chain in the mitochondria (Figure 4D). Notably, these processes are regulated by the mitochondria, which are the most important organelles for energy generation. Furthermore, the metabolism score for each cell was independently calculated, which also proved that malignant cells could play a crucial role in metabolic processes (Figure 4E–G). Finally, the enrichment analysis was conducted for comparison of the metabolic processes between the high- and low-MAM-score malignant cells. This indicated that the high-MAM-score malignant cells were positively correlated with the citrate (TCA) cycle, oxidative phosphorylation, mitochondrial ATP synthesis, NADH dehydrogenase activity, glycolysis/gluconeogenesis, etc. (Figure 4H). Together, the above results implied that MAM may depend on several mitochondrial metabolic processes, thus driving the malignant cells in HCC.

### 3.4. Identification of the Underlying Mechanisms of MAM

The “SCENIC” method employed to identify the underlying specific TFs of each cell type demonstrated that ELK3 (81 g) was higher in TEC, NR2F2 (11 g) was higher in CAF, STAT4 (14 g) was higher in T cells, and NFKB1 (58 g) was higher in TAM. CEBPA-extended (60 g) was higher in malignant cells, and BHLJE41-extended (12 g) was higher in B cells (Figure 5A). As previously mentioned, MAM scores in the non-immune cells, especially the malignant cells, were higher than those in the immune cells. Then, the activities of TFs were revealed by T-SNE, which demonstrated that the activities of TFs were higher in the high-MAM-score group than in the low-MAM-score group (Figure 5B–G). In addition, “RTN” was used to calculate the regulon scores of these TFs and examine whether these scores were different between the high- and low-MAM-score groups in the TCGA dataset. The results showed that most of the regulon scores were higher in the high-MAM-score group than in the low-MAM-score group (Figure 5H). Moreover, the cell-to-cell communication between the low- and high-MAM-score groups was tested, and the results indicated the enhanced strength of the high-MAM-score malignant cells and T cells (Figure 5I). In addition, the differences in the receptor ligands between low- or high-MAM-score malignant cells and other cell types implied the upregulation of multiple signaling pathways in the non-immune cell types, such as SPP1-CD44, and the downregulation in some immune cell types, such as MIF-(CD74 + CXCR4, Figure 5J,K). From the above results, we found that high-MAM-score malignant cells could influence the non-immune cell types and immune cell types, especially T cells, which revealed the role of MAM in influencing the tumor environment.

### 3.5. Differences in the Tumor Microenvironment Infiltration between Different MAM—Subgroups

To further determine the influence of TME on the HCC prognosis, CIBERSORT was used to estimate the fractions of each cell type of TME, which could prolong the prognosis of HCC (Appendix A). Three cell types (activated NK cells, resting memory CD4+ T cells, and CD8+ T cells) were selected, and it was speculated that these TME-related cell types could protect the normal cells from the invasion of tumor cells. We hypothesized that T cells and NK cells could be beneficial to the HCC prognosis. Notably, the CellChat analysis in this study indicated that malignant cells with high-MAM-score could transmit an increasing signal to T cells. Hence, the combined index considering both the MAM score and TME score was evaluated for predicting the prognosis of HCC patients. Further, Cox analysis was conducted to calculate the TME score of each sample, and the samples were divided into three subgroups, including the high MAM/low TME group, mixed group, and low MAM/ high TME group. KM analysis was conducted to evaluate the prognostic value between the three groups, and the results indicated that the high-MAM/low-TME-score group had shorter OS than other groups (Figure 6A,B; TCGA: *p* < 0.0001; ICGC: *p* = 0.034). The correlation between the combined marker and other classified markers was further investigated, indicating that the combined marker was closely related to other HCC subtypes, including *CTTNB1*, *LEE*, *Hoshida*, *Boyault*, and *Chiang* (Figure 6C). In addition, immune scores, stromal scores, scores of the immune checkpoints, and immune cell types were determined to elucidate the tumor immune infiltration landscape in the three groups. This indicated that though several immune checkpoints had no differences between the three groups, the immune-related points were higher in the high MAM/low TME group (Figure 6D). Mariathasan et al. established several tumor immune infiltration signatures [28]. According to these signatures, it was observed that the effector CD8+ T cell score was higher in the low MAM/high TME group and almost all the scores were higher in the high MAM/low TME group, except antigen processing machinery and EMT1, which is in agreement with the previous results (Figure 6E). Thus, combining the TME scores and MAM scores could better distinguish the tumor immune infiltration environment and improve the prognosis of HCC patients.

### 3.6. Mutation Landscape and Response to the Immune Checkpoint Block (ICB) Therapy

In this study, the landscape of genomic variations between the three heterogeneous clusters was represented, indicating that TP53 mutation was higher in the high MAM/low TME group (Figure 7A,B). TP53 is the most frequently mutated gene in HCC. Figure 7A shows several CNV, indicating that though there are no obvious differences in the amplification arm between the three groups, the mutation frequency of the depletion in the arm was much higher in the high MAM/low TME group, while much lower in the low MAM/high TME group. The TIDE platform demonstrated that the observed exclusion score was higher in the high MAM/low TME group, which is known to reflect the sensitivity to ICB (Figure 7C). Furthermore, two methods for predicting the response to ICB were applied. First, the IPS of CTLA and PD1 in the three groups was determined, which indicated that the sensitivity was higher in the low MAM/high TME group (Figure 7D). Then, the TIDE and submap platform were used to demonstrate the better response of the low MAM/high TME group to the ICB of PD1 (Figure 7E,F). Together, these results indicated that a combined index of the MAM and TME scores shows promising results for ICB, demonstrating that HCC patients in the low MAM/high TME group could better respond to ICB.

### 3.7. Response of the High- and Low-MAM-Score Groups to Chemotherapy

To date, TACE is widely used in clinical practice for HCC [29]. Hence, GSE104580 based on 66 samples from responders to TACE and 81 samples from non-responders to TACE was applied, and it was observed that the low-MAM-score group could better respond to TACE (*p* = 0.004, Figure 8A). The results implied that the MAM score could be a promising indicator to assess the suitability of a patient for chemotherapy. In addition, the PRISM dataset was used to identify the latent therapeutic drugs considering the CCLE data as a reference. The results indicated a positive correlation between lovastatin, fluvastatin, and simvastatin with low MAM (Figure 8B,C). It is well known that HMG-COA is the common target of the three compounds, and is involved in several metabolic pathways, such as cholesterol synthesis and oxidative stress [30]. When the expression of HMG-COA was analyzed in this study, HMG-COA expression levels were found to be higher in the high-MAM-score group (Figure 8E). Furthermore, the CMAP platform demonstrated that fasudil exerted more inhibitory effects on the high-MAM-score group when compared with the low-MAM-score group (Figure 8D). Fasudil is a potent Ca^2+^ channel antagonist and vasodilator [31]. Further, the expression of ATP2A2, which is a core gene to reflect the Ca pump and ER Ca^2+^ transportation, was investigated [32], and the results indicated that ATP2A2 was higher in the high-MAM-score group (Figure 8F). According to these results, we speculated HCC patients with a low MAM score may benefit from chemotherapy.

## 4. Discussion

MAM-related genes play crucial roles in the indirect system, which regulates the transmission of second messengers, such as Ca^2+^ [33]. The study results elucidated that the index based on the classification of these genes could predict the prognosis in the three datasets. In addition, large-scale transcriptome and microarray data proved the differences in the expression and stability of these genes. Generally, the MAM-related genes play crucial roles in the progression of HCC. Hence, a MAM-based score was constructed, which exhibited 5-year AUC values of 0.75 and 0.73 in the TCGA and ICGC cohorts, respectively, indicating the prognostic utility of this model. Five core genes (ACAT1, PACS2, VDAC1, MFN1, and ATAD3A) playing critical roles in cancer were identified in this study. ACAT1 is recognized as a tumor suppressor and is involved in several mitochondrial-induced metabolic pathways. In addition, it has also been noted in the literature that ACAT1 is highly expressed in lung cancer and prostate cancer. We speculated that this may be caused by the strong heterogeneity of tumors, which also needs to be verified in our follow-up experiments [34,35]. Deng et al. observed that PACS2 could regulate MAM to impair erectile function in rats with prostate cancer [36]. VDAC1 is one of the major Ca^2+^ transport channels in the mitochondria. MAM regulates the transfer of Ca^2+^ from the ER to mitochondria via the VDAC1-GRP75-IP3R3 macromolecular complex [37]. In addition, Azeez et al. observed that VDAC1 could also mediate progesterone-triggered-Ca^2+^ in breast cancer [38]. Li et al. believed that the MFN1/MFN2 pathway could promote ferroptosis through MAM in pancreatic cancer [39]. At the same time, Lang et al. reviewed several cancer-related studies on ATAD3A and concluded that ATAD3A could control the progression of cancers by regulating the mitochondrial dynamics and ER [40]. The findings indicated that these genes could regulate the MAM to control the mitochondrial functions, thus promoting cancer progression. Therefore, it was speculated that the MAM score could indicate tumor progression and contribute to the prediction of prognosis in HCC patients.

Due to the high heterogeneity in HCC, a single clinical index could not reflect the complexity of TME. As a result, drug tolerance and immune exclusion were observed post chemotherapy or immune therapy. The development of a single-cell technology led to the identification of the landscape of TME. The AUCell analysis indicated that the MAM score was higher in the non-immune cell types, especially malignant cells, than the immune cells. Additionally, malignant cells exhibited more metabolism characteristics. In particular, the energy metabolism pathways were dependent on the mitochondrial functions, which further proved our approach. Additionally, drug sensitivity analysis revealed the potential drugs, including statins and fasudil, for HCC patients with low MAM scores. Notably, these drugs are individually used to reduce cholesterol and Ca^2+^ levels. Aleix and co-workers found that CAV1 knockout results in the low stability of ER-mitochondrial contact sites, which led to the accumulation of free cholesterol in MAMs [41]. This implied that the control of metabolism by the underlying compounds could help in a better prognosis of HCC. Except for Ca^2+^ signaling, other energy signals and kinases were also observed to be active in malignant cells. Notably, the tumor cells are more likely to go through the state of cellular senescence for defense against cancer. During this period, ROS from MAM impacts the energy balance of mitochondria, thus activating the AMPK signaling pathway, Ca^2+^ overload, and mitochondrial dysfunction [42]. In addition, the NAD/NADH ratio is also an energy indicator to assess the stability of MAM tethering [18]. Thus, controlling the distance between mitochondria and ER is vital for the energy metabolism of cancer cells. In particular, the classification based on the MAM-related genes and other HCC subtypes was compared, which indicated that the MAM subtype was closely related to other subtypes. Hence, it was speculated that similar to the MAM subtype, these subtypes may also be involved in several pathways of energy metabolism. However, over the past few years, research focused on the second messengers, while ignoring the subcellular organelles, which functioned to transmit these signals. To date, most of the research has concentrated on the organelle functions, such as lysosomes, ribosomes, etc. [43,44]. Thus, the results from this study verified that MAM may influence energy metabolism, thereby promoting HCC progression.

The results from the CellChat analysis demonstrated that the malignant cells delivered enhanced signals to T cells when compared to other cell types. In addition, the co-interaction strength was higher in the high-MAM-score malignant cells than in the low-MAM-score malignant cells. In addition, the AUCell analysis indicated that the MAM score was higher in the malignant cells and relatively lower in the T cells. Hence, it was concluded that the antagonism of T cells and high-MAM-score malignant cells may decide the destiny of HCC patients. Then, CIBERSORT was used to identify three immune cell types (activated NK cells, resting memory CD4+ T cells, and CD8+ T cells), which could prolong the prognosis of HCC patients. It is well known that NK and CD8+ T cells could specifically kill the tumor cells through several common pathways, including perforin, granzyme, Fas/FasL, TNF-α, and IFN-γ. In addition, CD4+ T cells could assist CD8+ T cells to simultaneously participate in cellular immune response as well as clear cell tumors [45,46]. Based on the evidence, it was speculated that an immune-related prognostic signature could be better to distinguish the TME and predict the prognosis of HCC patients. As expected, HCC patients with a high MAM score/low TME score tend to have a worse prognosis and higher tumor immune infiltration signatures. To date, the major limitation of CART therapy in solid tumors is CD8+ T cell exhaustion. The CD8+ T cells that lose their functional capabilities express a high level of PD-1, thus shaping the TME and promoting cancer progression [47]. Hence, further investigation on the correlation between CD8+ T cell exhaustion and MAM score could help us better understand the regulatory mechanisms of TME. Briefly, a combination of the MAM score and TME score could be a promising marker for predicting the survival rates of HCC. 

In addition to the above results, genome somatic mutations and CNV also revealed the heterogeneity of HCC [48]. The gene mutations can directly affect the TME by recruitment or exclusion of immune cell penetration. Some studies have demonstrated that cancer cells in HCC patients with TP53 mutations tend to escape from immune therapy, and HCC patients with CCTNB1 mutations are more likely to show immune tolerance and exclusion [49,50]. Jérôme et al. observed that TP53 mutation status was also related to TME [51]. It has also been demonstrated that the immunogenicity of TP53 mutations is driven by complex dynamics, including the affinity of T cell antigen receptor (TCR), transport of T cells to TME, etc. [52]. Notably, the study results indicated high TP53 and CCNTB1 mutation frequency, especially the TP53 mutation, in the high-MAM-score/low-TME-score group. In addition, the alterations in CNV had a larger contribution to immune therapy than somatic mutations. In this study, the results revealed that patients in the high-MAM-score/low-TME-score group tend to acquire a high frequency of depletion in the arm, thereby indicating that the patients in the high-MAM-score/low-TME-score group may be insensitive to immune therapy. As expected, the patients in the low-MAM-score/high-TME-score group have a better response to PD-1 immune therapy. In recent years, immune checkpoint inhibitors have brought fundamental changes to the treatment of HCC. PD-1 is an immune checkpoint molecule, and the latest clinical trials on HCC have shown unprecedented success in immunotherapy targeting the PD-1/PD-L1 axis [53,54,55]. However, it still faces great challenges, and its low remission rate is yet to be solved. For most HCC patients, the PD-1/PD-L1 pathway is not the only limiting factor of antitumor immunity, and blocking only the PD-1/PD-L1 axis is not enough to stimulate an effective antitumor immune response. After first-line immunotherapy combination treatment, second-line targeted treatment is a viable option. Notably, Yu and co-workers discovered that disturbed mitochondrial dynamics reinforce T cell exhaustion and PD1 to reconstruct TME [56]. Hence, it can be speculated that the maintenance of the MAM could help stabilize the mitochondria and strengthen the anti-tumor immune system. In general, this study showed that a combination of the MAM score and TME score could be a promising marker for predicting the response of HCC patients to ICB therapy. 

In summary, this study is the first investigation of MAM in HCC considering publicly available data. As an organelle that transmits energy, MAM plays a crucial role in the HCC progression. The results from this study also showed that the MAM score could be used to identify the suitability of HCC patients for chemotherapy. It is well known that different molecular characteristics usually reflect different pathological features. Due to the high heterogeneity of HCC, a single index cannot reflect the differences in the tumor immune infiltration. Therefore, the combination of the TME score and MAM score considered in this study could be a better solution to distinguish TME and predict the prognosis of HCC. However, there are some limitations to this approach that should be considered. First, a few GEO datasets including the detailed prognosis information do not detect the expression of some MAM-associated genes, which restricts us to expand our patient cohorts. Second, all the patients enrolled in this research were retrospective, and a prospective study should be considered for validation of the results. Nevertheless, this research showed that the MAM score reflected the metabolic characteristics, and combining the MAM score with the TME score could better predict the prognosis of HCC patients. 

## 5. Conclusions

MAM score is a promising index for determining the need for chemotherapy, which reflects the energy metabolic pathways. A combination of the MAM score and TME score could be a better indicator to predict prognosis and response to immune therapy.

## Figures and Tables

**Figure 1 jcm-12-01830-f001:**
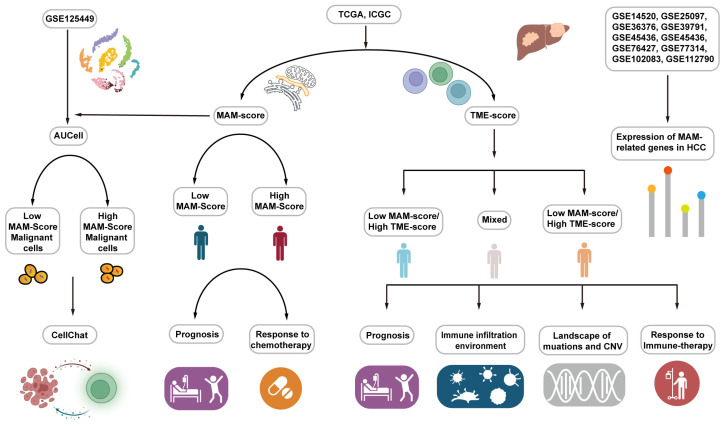
Overview of the workflow. HCC datasets (TCGA, ICGC, GSE14520, GSE25097, GSE36376, GSE39791, GSE45436, GSE54236, GSE76427, GSE77314, GSE102083, and GSE112790) were downloaded to demonstrate the differences in the MAM-associated genes between the normal and HCC groups. MAM score was constructed using the TCGA dataset and validated using the ICGC dataset. Additionally, the MAM score facilitated the prognosis as well as chemotherapy response. GSE125449 single-cell data were used to indicate the TME of HCC. In addition, AUCell analysis was used for detecting the ROC of MAM. Furthermore, CellChat analysis was used for comparing the co-communication of different cell types. Then, TME score was constructed and the samples were classified into three groups based on the MAM and TME scores. Finally, the prognosis, immune infiltration environment, landscape of mutation and CNV, and response to immune therapy of the three groups were determined.

**Figure 2 jcm-12-01830-f002:**
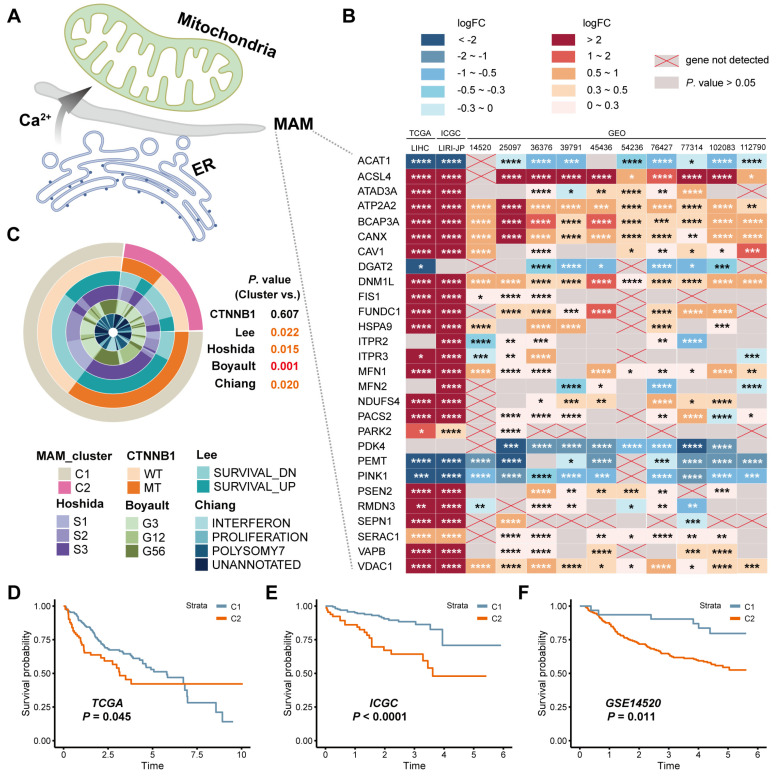
Identification of the expression and prognostic value of MAM in HCC. (**A**): Illustration of the MAM structure. (**B**): Heatmap of the expression of MAM-related genes in several HCC datasets. (**C**): Correlations between the generated cluster and other HCC subtypes. (**D**–**F**): KM curves of different MAM-related clusters in the TCGA, ICGC, and GSE14520 datasets. * *p* < 0.05; ** *p* ≤ 0.01; *** *p* ≤ 0.001; and **** *p* ≤ 0.0001.

**Figure 3 jcm-12-01830-f003:**
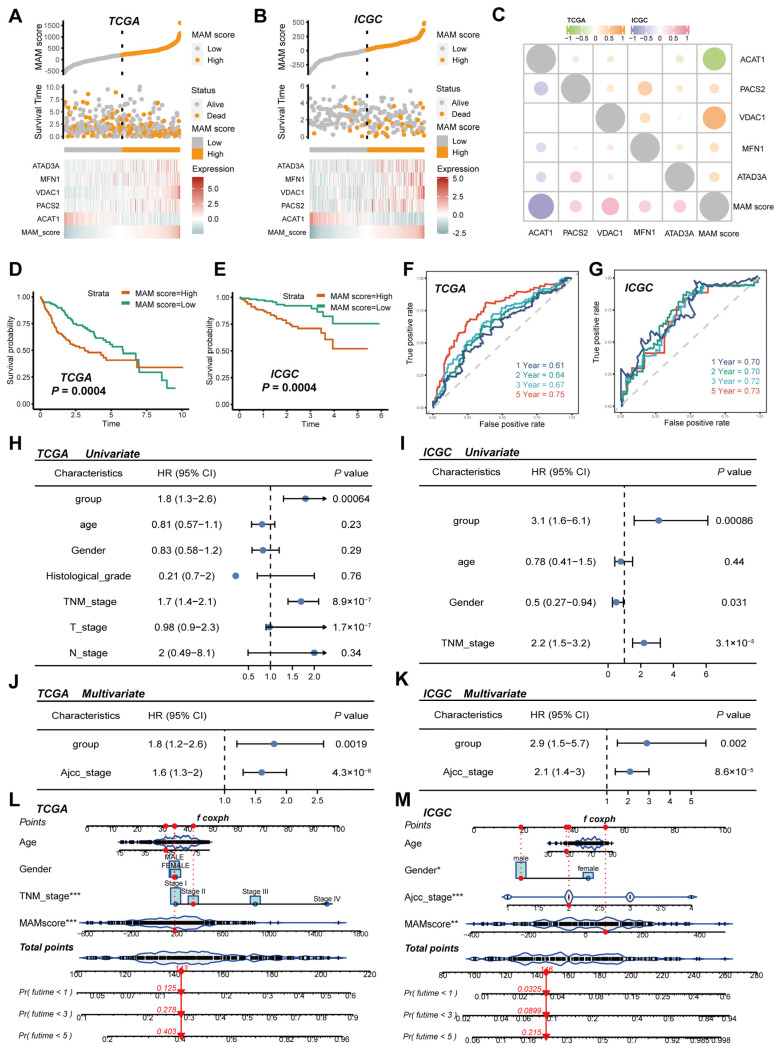
Construction and verification of the MAM scores in HCC. (**A**,**B**): Distribution of the MAM scores and survival status of HCC patients in the TCGA and ICGC datasets. (**C**): Expression correlation between the MAM score and its constitutive genes. (**D**,**E**): KM curves of the low- and high-MAM−score groups in the TCGA and ICGC datasets. (**F**,**G**): Time-dependent ROC analysis of the TCGA and ICGC datasets for MAM scores at 1, 3, 4, and 5 years. (**H**–**K**): Univariate Cox regression and multivariate Cox regression analyses for MAM scores and other clinical characteristics in the TCGA and ICGC datasets. (**L**,**M**) Nomogram predicts 1-, 3-, and 5-year OS of HCC samples based on the TCGA and ICGC datasets. * *p* < 0.05; ** *p* ≤ 0.01; *** *p* ≤ 0.001.

**Figure 4 jcm-12-01830-f004:**
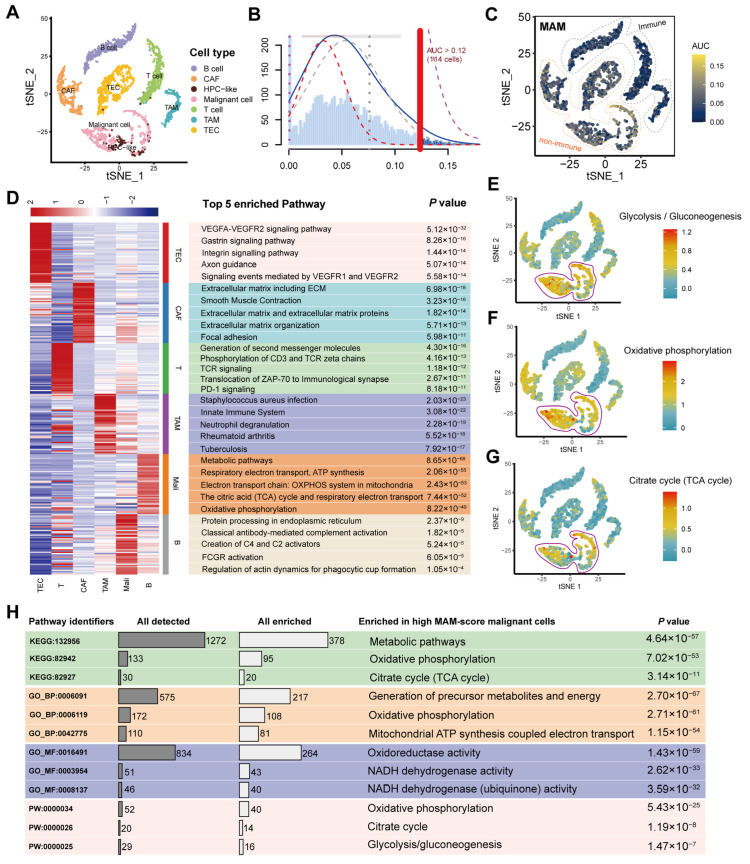
Identification of the MAM score and its potential enriched pathways at the single-cell level. (**A**): T-SNE visualization of various cell types. (**B**): Frequency of calculating the MAM scores by AUCell analysis. (**C**): AUC of MAM score visualized by T-SNE. (**D**): Top 5 enrichment pathways of different cell types. (**E**–**G**): Metabolism scores of different cell types by metabolism. (**H**): Comparison of the metabolic pathway enrichment in the high-MAM-score malignant cells and low-MAM-score malignant cells.

**Figure 5 jcm-12-01830-f005:**
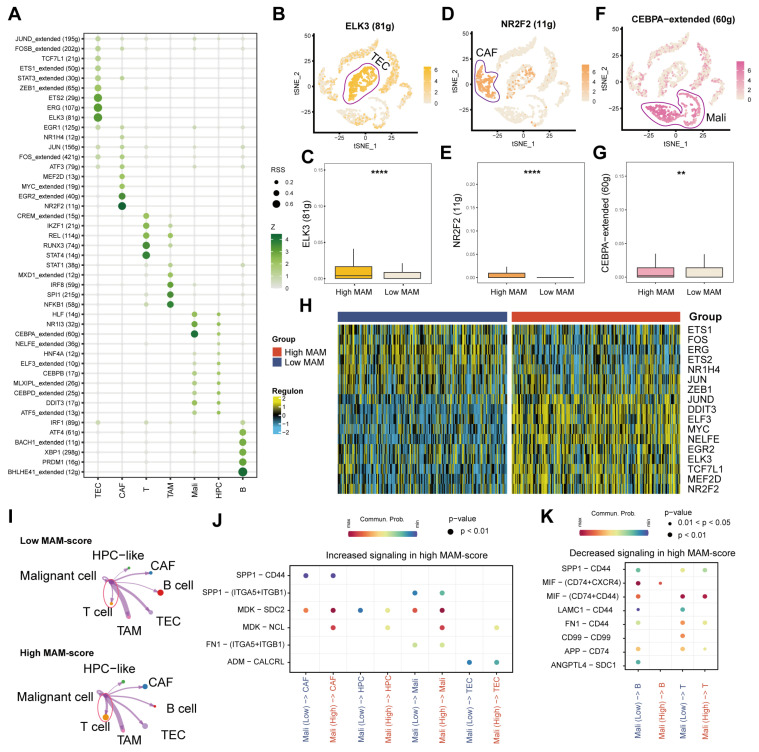
Identification of the mechanisms underlying MAM. (**A**): Activity of TFs of different cell types. (**B**–**G**): Boxplots comparing ELK3 (81 g), NR2F2 (11 g), and CEBPA-extended (60 g) expressions of TFs between the low- and high-MAM-score groups. (**H**): Heatmap indicating the specific regulon activity profiles of HCC between the low- and high-MAM-score groups. (**I**) Differences in the interaction strengths between the low-MAM-score malignant cells, high-MAM-score malignant cells, and other cell types. (**J**,**K**): Increasing and decreasing receptor–ligand signaling patterns between the high- and low-MAM-score malignant cells. ** *p* ≤ 0.01; **** *p* ≤ 0.0001.

**Figure 6 jcm-12-01830-f006:**
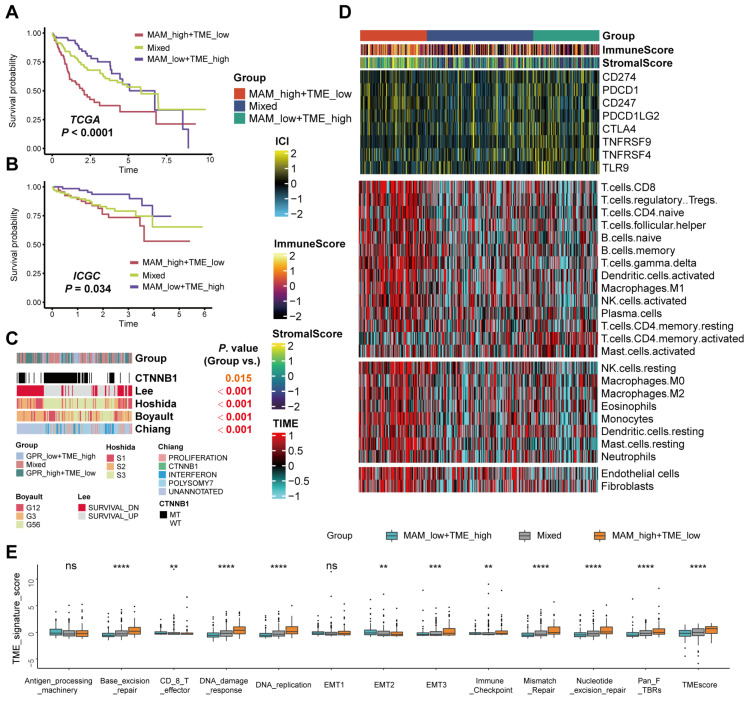
Differences in the tumor microenvironment infiltration between different MAM–TME subgroups. (**A**,**B**): KM curves of low-MAM/high-TME-score group, mixed group, and high MAM/low TME group in the TCGA and ICGC datasets. (**C**): Correlations between the combined subtypes and other HCC subtypes. (**D**): Heatmap exhibiting the immune score, stromal score, and several cell types calculated through CIBERSORT analysis of the combined subtypes. (**E**): Tumor immune infiltration signatures of the combined subtypes. ** *p* ≤ 0.01; *** p ≤ 0.001; **** *p* ≤ 0.0001; ns, not significant.

**Figure 7 jcm-12-01830-f007:**
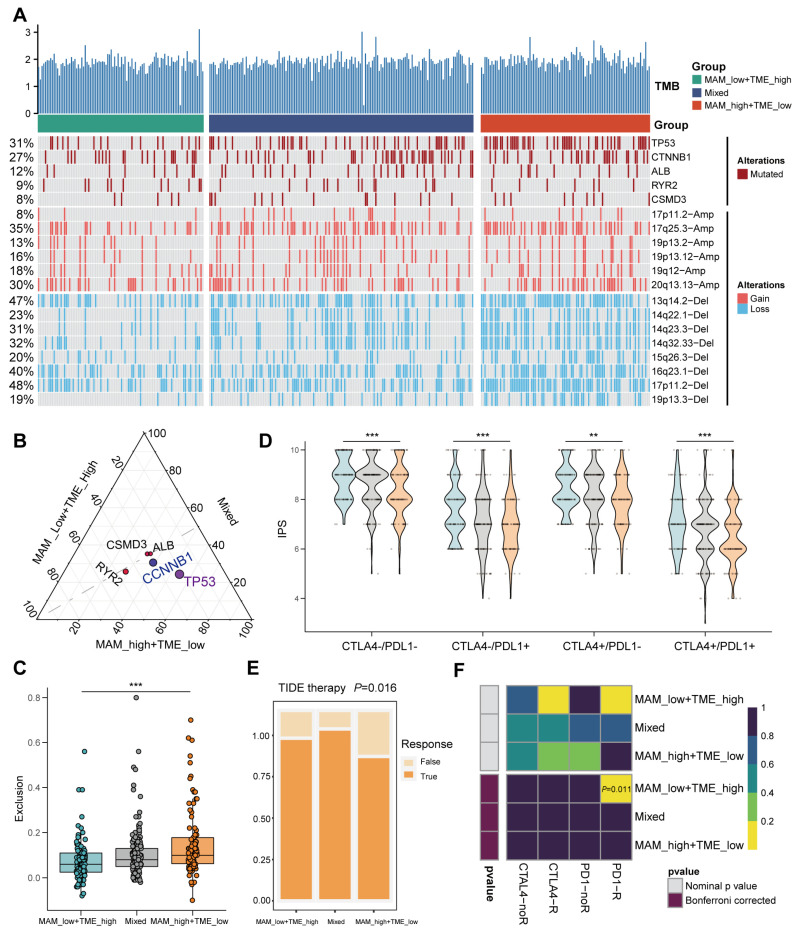
Mutation landscape and response to ICB. (**A**): Landscape of genomic mutations and CNV data of HCC in the low MAM/high TME group, mixed group, and high MAM/low TME group in the TCGA dataset. (**B**): Triangle plot and prominent dots of the top five mutated frequencies between the combined groups. (**C**): Boxplot of exclusion between the combined groups in the TIDE platform. (**D**): Relative probability of the effectiveness of anti-PD-1 and anti-CTLA-4 therapies in patients with HCC in the combined groups. (**E**): Proportion of HCC patients with TIDE response of the combined groups. (**F**): Combined analysis of TIDE and submap to predict the response of the combined groups to ICB therapy. ** *p* ≤ 0.01; *** *p* ≤ 0.001.

**Figure 8 jcm-12-01830-f008:**
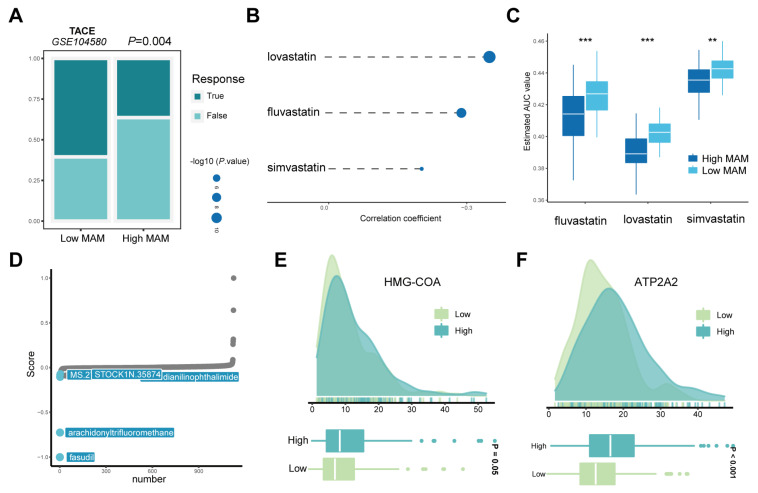
Differences in the response to chemotherapy between the high- and low-MAM-score groups. (**A**) Proportion of HCC patients with response to TACE therapy in the low- and high-MAM-score groups. (**B**,**C**): Spearman’s correlation analysis and differential drug response in the low- and high-MAM-score groups. (**D**): CMAP analysis of computational drug repositioning in the low- and high-MAM-score groups. (**E**,**F**): Ridge and box plots of HMG-COA and ATP2A2 in the low- and high-MAM-score groups. ** *p* ≤ 0.01; *** *p* ≤ 0.001.

## Data Availability

The results of this study are supported by the data available from the TCGA “https://portal.gdc.cancer.gov (accessed on 12 January 2023)”, ICGC “https://dcc.icgc.org/ (accessed on 12 January 2023)”, and GEO “https://www.ncbi.nlm.nih.gov/geo/ (accessed on 12 January 2023)”.

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
