# Peer review of "Mitochondria-Associated Endoplasmic Reticulum Membrane (MAM) Is a Promising Signature to Predict Prognosis and Therapies for Hepatocellular Carcinoma (HCC)"

_jcm, 2023, doi:10.3390/jcm12051830_

Round 1

Reviewer 1 Report

Chen et al., in their paper entitled “Mitochondria-associated endoplasmic reticulum membrane (MAM) is a promising signature to predict prognosis and therapies for hepatocellular carcinoma (HCC)” have studied the expression of MAM-related genes in public datasets. The single-cell sequencing studies carried out by the authors further validates the prognostic significance of MAM in HCC.

However, the authors are required to address the following queries:

1.       SELENOM (Selenoprotein M) has been implicated in cancer, with evidence suggesting that it may play a role in hepatocellular carcinoma (HCC). Studies have shown elevated levels of SELENOM expression in HCC cell lines as compared to normal hepatocytes. Previous investigation confirmed that findings in tumor samples from HCC patients, suggesting that SELENOM expression could be a potential prognostic tool for understanding HCC progression. SELENOM has also been found to be a positive regulator of the Thioredoxin reductase (TXN) system, which plays a critical role in regulating cell growth and survival. SELENOM has the potential to affect both mitochondrial calcium levels and the morphology of MAMs (Mitochondria-associated endoplasmic reticulum membrane). Its interaction with calcium channels and pumps such as SERCA and IP3R may result in an increase of calcium flux to the mitochondria, potentially regulating ATP generation and ROS generation. We suggest that Selenoprotein M may play a role in the development and progression of Human Hepatocellular carcinoma and regulate the mitochondria associated endoplasmic reticulum membrane (MAMs). Hence, SELENOM could be a potential marker to follow the progression of Human hepatocellular carcinoma (HCC) cancer.

2.       Why Selenoprotein M (SELENOM) highly expressed in hepatocellular carcinoma compared to normal cells?

3.       Whether your single cell data analysis can provide insights on the impact of Selenoprotein M (SELENOM) on MAMs?

4.       Further, investigation is necessary to analyze the role of Selenoprotein M in MAMs.

5.       Line 26: Provide the full form of AUCell.

6.       Line 103: The word “divide” may be replaced with more appropriate word such as “classify” or other such word.

7.       Line 110: The space between “GSE” and “112790” should be removed as is interpreted from the other such words.

8.       Line 112 and 113: The sentence should be restructured and may be written as “the MAM score facilitated the prognosis as well as chemotherapy response.

9.       Line 134: The phrase “R package TCGA biolinks” seems to be errorful due to the word editor. Please do the needful.

10.   Line 142: shu should be written as Shu

11.   The name of R package should be written in “” wherever applicable.

12.   Line 252: Finally, five genes, including ACAT1, PACS2, VDAC1, MFN1, and ATAD3A, were identified. Here in this sentence, the use of word “including” is redundant.

13.   Line 253: In addition to the TCGA dataset as a training set, the ICGC dataset was also used as a validation set. Here in this sentence, ambiguity exists. As the sentence is written, it may be that the author says “The TCGA dataset was used both as training set and validation set. The ICGC dataset was used as validation set”.

14.   As per the Abstract, the author has used the TCGA dataset as training dataset while the ICGC dataset as validation set. However, it seems that the sentence is not providing this information or may confuse the readers. Please correct the sentence accordingly.

15.   Line 256 and 257: The words “In addition” and “also” are same and one of them is redundant. Please correct the sentence accordingly.

16.   Figure 3: The legends are visible till panel 3C while 3D and 3E are visible but no legend. It appears that it has happened during journal processing or the authors have forgotten to put it. Please do the needful.

17.   Line 278-283: The missing legends for Figure 3 for panel 3D-K are present here. Please edit the raw manuscript so that it does not happen like this. Please recheck it before final submission on the journal webpage.

18.   Reviewers are unable to locate the legends for panel 3L and 3M. Please add them or locate them and do proper editing.

19.   All the packages of R should be mentioned as R package “name of package” wherever applicable for better readability of readers.

20.   Figure 4: The legends are visible till panel 4F. It looks like that the authors have forgotten to put it. Please do the needful.

21.   Line 379: The acronym ICB is not suitable for the phrase Immune checkpoint therapy. Either add a suitable acronym such as ICT or change the full form.

22.   X-axis, Y-axis and other labels in the panel of Figures should be properly written. For instance, Figure 8, some are written in small letters while some are written with first letter capitalized. It is suggested that some pattern should be followed throughout the paper.

23.   Please follow uniform pattern of writing the abbreviations such as first letter capitalized or all letters small for most cases. 

Author Response

General Comment: Chen et al., in their paper entitled “Mitochondria-associated endoplasmic reticulum membrane (MAM) is a promising signature to predict prognosis and therapies for hepatocellular carcinoma (HCC)” have studied the expression of MAM-related genes in public datasets. The single-cell sequencing studies carried out by the authors further validates the prognostic significance of MAM in HCC

Response: We appreciate your comments of our work. In response to the issues, we have made the following revisions.

Comment 1: SELENOM (Selenoprotein M) has been implicated in cancer, with evidence suggesting that it may play a role in hepatocellular carcinoma (HCC). Studies have shown elevated levels of SELENOM expression in HCC cell lines as compared to normal hepatocytes. Previous investigation confirmed that findings in tumor samples from HCC patients, suggesting that SELENOM expression could be a potential prognostic tool for understanding HCC progression. SELENOM has also been found to be a positive regulator of the Thioredoxin reductase (TXN) system, which plays a critical role in regulating cell growth and survival. SELENOM has the potential to affect both mitochondrial calcium levels and the morphology of MAMs (Mitochondria-associated endoplasmic reticulum membrane). Its interaction with calcium channels and pumps such as SERCA and IP3R may result in an increase of calcium flux to the mitochondria, potentially regulating ATP generation and ROS generation. We suggest that Selenoprotein M may play a role in the development and progression of Human Hepatocellular carcinoma and regulate the mitochondria associated endoplasmic reticulum membrane (MAMs). Hence, SELENOM could be a potential marker to follow the progression of Human hepatocellular carcinoma (HCC) cancer.

Response: Thank you very much for your suggestion. Actually, the research on MAM in HCC is still unclear, and we are exploring other molecules that may affect MAM in other studies. By reviewing the literature, we have found that SELENOM indeed affects calcium and mitochondria regulation pathways, which provides us with a new perspective. We will further study Selenoprotein M in the subsequent research.

Comment 2: Why Selenoprotein M (SELENOM) highly expressed in hepatocellular carcinoma compared to normal cells?

Response: We found that SELENOM is highly expressed in HCC and is closely related to tumor grading through the relating literatures, but there are few articles related to the mechanism. We speculated that this is because the transcription factors which can bind to SELENOM are highly expressed in HCC.

Comment 3: Whether your single cell data analysis can provide insights on the impact of Selenoprotein M (SELENOM) on MAMs?

Response: Thank you very much for your suggestion. We detected the expression of SELENOM in the single-cell data. Unfortunately, SELENOM was not detected in this dataset. This situation often occurs because the 10x single-cell technology has limitations in terms of detection depth and coverage, and typically detects only about 3,000-5,000 coding RNAs, which represent one-fifth of all coding RNAs. However, we believe the potential role of SELENOM in HCC, and we will directly validate its expression and function in further experiments.

Comment 4: Further, investigation is necessary to analyze the role of Selenoprotein M in MAMs.

Response: Thank you very much for your suggestion. In subsequent research, we will further explore the role of Selenoprotein M in MAM.

Comment 5: Line 26: Provide the full form of AUCell.

Response: Thank you very much for your attention. The full form of AUCell is “Assessing the Uncertainty of Clustering in single-cell RNA-seq data using a gene co-expression network” and we have revised it (Revised in Methods, line 27-28; 163-164).

 Comment 6: Line 103: The word “divide” may be replaced with more appropriate word such as “classify” or other such word.

Response: Thank you very much for your suggestion. we have replaced the word with “classify” (Revised in Methods, line 224).

Comment 7: Line 110: The space between “GSE” and “112790” should be removed as is interpreted from the other such words.

Response: Thanks for your attention. It is our negligence caused this error. we have removed the space (Revised in Methods, line 232).

Comment 8: Line 112 and 113: The sentence should be restructured and may be written as “the MAM score facilitated the prognosis as well as chemotherapy response.

Response: Thanks for your suggestion. we have replaced the sentence with which you suggested (Revised in Methods, line 234-235).

Comment 9: Line 134: The phrase “R package TCGA biolinks” seems to be errorful due to the word editor. Please do the needful.

Response: We felt sorry for our negligence. There was no space between TCGA and biolinks. Thanks for your attention, we have revised it (Revised in Methods, line122). In addition, the detailed information about “TCGAbiolinks” could be enquired in the website(http://www.bioconductor.org/packages/release/bioc/html/TCGAbiolinks.html) .

Comment 10: Line 142: shu should be written as Shu.

Response: Thanks for your attention, we have replaced the word with “Shu”(Revised in Methods, line 131).

Comment 11: The name of R package should be written in “” wherever applicable.

Response: Thanks for your attention, we felt sorry for our negligence. we have revised these errors (Revised in Methods, line 122; 136).

Comment 12: Line 252: Finally, five genes, including ACAT1, PACS2, VDAC1, MFN1, and ATAD3A, were identified. Here in this sentence, the use of word “including” is redundant.

Response: Thanks for your suggestion. we have revised this error (Revised in Results, line 268).

Comment 13: Line 253: In addition to the TCGA dataset as a training set, the ICGC dataset was also used as a validation set. Here in this sentence, ambiguity exists. As the sentence is written, it may be that the author says “The TCGA dataset was used both as training set and validation set. The ICGC dataset was used as validation set”.

Response: Thanks for your suggestion. In our study, TCGA dataset was used as a training set and ICGC dataset was used a validation set. We have revised this error (Revised in Results, line 269-270).

Comment 14: As per the Abstract, the author has used the TCGA dataset as training dataset while the ICGC dataset as validation set. However, it seems that the sentence is not providing this information or may confuse the readers. Please correct the sentence accordingly.

Response: Thanks for your suggestion. In our study, TCGA dataset was only used as a training set and ICGC dataset was only used a validation set. We have revised this sentence into “The TCGA-LIHC dataset was only used as a training set. In addition, ICGC and several GEO datasets were used for validation.” (Revised in Results, line 24-26)

Comment 15: Line 256 and 257: The words “In addition” and “also” are same and one of them is redundant. Please correct the sentence accordingly.

Response: Thanks for your suggestion. We deleted the word “also” in this sentence (Revised in Results, line 272).

Comment 16: Figure 3: The legends are visible till panel 3C while 3D and 3E are visible but no legend. It appears that it has happened during journal processing or the authors have forgotten to put it. Please do the needful.

Response: We speculated it could happen during the journal processing. We hope you could see the legends in the revised manuscript. The whole figure legends described: Construction and verification of the MAM scores in HCC. (A-B): Distribution of the MAM scores and survival status of HCC patients in the TCGA and ICGC datasets. (C): Expression correlation between the MAM score and its constitutive genes. (D-E):KM curves of the low and high MAM-score groups in the TCGA and ICGC datasets. (F-G): Time-dependent ROC analysis of the TCGA and ICGC databases for MAM scores at 1, 3, 4, and 5 years. (H-K): Univariate Cox regression and multivariate Cox regression analyses for MAM scores and other clinical characteristics in the TCGA and ICGC datasets. (L-M) Nomogram predicts 1-, 3-, and 5-year OS of HCC samples based on the TCGA and ICGC datasets (Revised in Results, line 291-298).

Comment 17: Line 278-283: The missing legends for Figure 3 for panel 3D-K are present here. Please edit the raw manuscript so that it does not happen like this. Please recheck it before final submission on the journal webpage.

Response: We have rearranged the edited docx file provided by JCM journal, and we apologized for the mistake. We hope you can see the correct order in the latest version (Revised in Results, line 291-298).

Comment 18: Reviewers are unable to locate the legends for panel 3L and 3M. Please add them or locate them and do proper editing.

Response: We have rearranged the edited docx file provided by JCM journal, and we apologized for the mistake. We hope you can see the correct order in the latest version (Revised in Results, line 291-298).

Comment 19: All the packages of R should be mentioned as R package “name of package” wherever applicable for better readability of readers.

Response: Thanks for your attention, we felt sorry for our negligence. we have revised these errors (Revised in Methods, line 173).

Comment 20: Figure 4: The legends are visible till panel 4F. It looks like that the authors have forgotten to put it. Please do the needful.

Response: We speculated that it could happen journal processing. And now we have rearranged the edited docx file provided by JCM journal. We hope you can see the correct order in the latest version (Revised in Results, line 323-328).

The whole figure legends described: Identification of the MAM score and its potential enriched pathways at the single-cell level. (A): t-SNE visualization of various cell types. (B): Frequency of calculating the MAM scores by AUCell analysis. (C): AUC of MAM score visualized by t-SNE. (D): Top 5 enrichment pathways of different cell types. (E-G): Metabolism scores of different cell types by scmetabolism. (H): Comparison of the metabolic pathway enrichment in the high MAM-score malignant cells and low MAM-score malignant cells.

Comment 21: Line 379: The acronym ICB is not suitable for the phrase Immune checkpoint therapy. Either add a suitable acronym such as ICT or change the full form.

Response: Thanks for your attention, we neglected the word “block”. The full form is revised as “immune checkpoint block (ICB) therapy” (Revised in Results, line 394).

Comment 22: X-axis, Y-axis and other labels in the panel of Figures should be properly written. For instance, Figure 8, some are written in small letters while some are written with first letter capitalized. It is suggested that some pattern should be followed throughout the paper.

Response: Thanks for your attention, we have revised these errors (Revised in Results, line 387; 438).

 Comment 23: Please follow uniform pattern of writing the abbreviations such as first letter capitalized or all letters small for most cases.

Response: Thanks for your attention, we have revised these errors (Revised in line 157, 173; 302; 324; 325; 336; 603).

Reviewer 2 Report

Thank you very much for the opportunity to review this important manuscript. The manuscript evaluated a vital topic and had scientific soundness and a good methodology. However, I have several comments.

The manuscript needs thorough language and grammar editing. The manuscript has several typos, grammar errors, as well as inappropriate use of terminology. To name a few:

-Page 2, line 55, “lenvartini”

-Page 2, line 58, “chemotherapy resistance” when talking about sorafenib and lenvatinib

The manuscript's aim should be better clarified in the introduction. Whether it was the early diagnosis and HCC prognosis prediction?

Please remove the statements related to methods and summary of the results from the last paragraph of the introduction. I advise the authors to conclude the last paragraph of the introduction with the study hypothesis and aims. Additionally, please move the figure-1 to the methods.

A similar problem exists in the results section, and several statements including comments are present in the results, while these statements should be found in the discussion. For example, page 6, lines 241-243, “Together, these results 241 speculated that the MAM-related gene signature could play a vital role in the progression 242 of HCC and influence the prognosis of HCC patients.” and page 11, lines 332-334 “From the above results, it was speculated that the 332 high MAM-score malignant cells could influence the non-immune cell types and immune 333 cell types, especially T cells, by multiple signaling pathways simultaneously.” Additionally, the statement on page 15, lines 421-423, is a very far-fetched statement and contradicts common knowledge of relative chemotherapy inefficiency in HCC (“According to these results, it was speculated that the application of personalized chemotherapy based on the MAM score would lead to curative effects.” ).

Please discuss the study results in light of the recent clinical trials in HCC. The immunotherapy-based combinations became the standard of care in the treatment of advanced HCC, and I recommend the authors comment on the possible use of the MAM score in patients treated with immunotherapy. Several recent papers summarizing the recent HCC clinical trials could be discussed in this regard (PMID: 36533987, PMID: 36601120, PMID: 36497349).

Author Response

General Comment: Thank you very much for the opportunity to review this important manuscript. The manuscript evaluated a vital topic and had scientific soundness and a good methodology. However, I have several comments.

Response: We appreciate the reviewer’s recognition of our work. In response to the issues, we have made the following revisions.

Comment 1: The manuscript needs thorough language and grammar editing. The manuscript has several typos, grammar errors, as well as inappropriate use of terminology. To name a few:

Page 2, line 55, “lenvartini”

Page 2, line 58, “chemotherapy resistance” when talking about sorafenib and Lenvatinib

Response: Thanks for your suggestion, we have revised these errors (Revised in Introduction, line 57). In addition, we asked for Bullet Edits Limited to review our manuscript thoroughly again.

Comment 2: The manuscript's aim should be better clarified in the introduction. Whether it was the early diagnosis and HCC prognosis prediction?

Response: Thanks for your suggestion, we have revised this paragraph (Revised in Introduction, line 102-107).

Comment 3: Please remove the statements related to methods and summary of the results from the last paragraph of the introduction. I advise the authors to conclude the last paragraph of the introduction with the study hypothesis and aims. Additionally, please move the figure-1 to the methods.

Response: Thanks for your suggestion, we have moved this part into the methods (Revised in Methods, line 217-141). In addition, the last paragraph of the introduction was revised (Revised in Introduction, line 102-107).

Comment 4: A similar problem exists in the results section, and several statements including comments are present in the results, while these statements should be found in the discussion. For example, page 6, lines 241-243, “Together, these results 241 speculated that the MAM-related gene signature could play a vital role in the progression 242 of HCC and influence the prognosis of HCC patients.” and page 11, lines 332-334 “From the above results, it was speculated that the 332 high MAM-score malignant cells could influence the non-immune cell types and immune 333 cell types, especially T cells, by multiple signaling pathways simultaneously.” Additionally, the statement on page 15, lines 421-423, is a very far-fetched statement and contradicts common knowledge of relative chemotherapy inefficiency in HCC (“According to these results, it was speculated that the application of personalized chemotherapy based on the MAM score would lead to curative effects.” ).

Response: Thanks for your suggestion. We are so sorry that we used too many vague words in the results section. Your comments are very helpful to improve the quality of our article. These parts were revised (Revised in Results, line 257-259; 347-349; 436-437).

Comment 5: Please discuss the study results in light of the recent clinical trials in HCC. The immunotherapy-based combinations became the standard of care in the treatment of advanced HCC, and I recommend the authors comment on the possible use of the MAM score in patients treated with immunotherapy. Several recent papers summarizing the recent HCC clinical trials could be discussed in this regard (PMID: 36533987, PMID: 36601120, PMID: 36497349).

Response: Thanks for your suggestion. In response to your suggestion, we have summarized and discussed the current status of immunotherapy in HCC, and relevant sections have been modified in the discussion part (Revised in Discussion, line 538-546).

Reviewer 3 Report

Chen et al performed an interesting study in which they developed  the mitochondria-associated endosplaamic reticulum membrane (MAM) score. This score, associated with newly constructed tumor-microenvironment score, was shown to determine the prognosis of patients with hepatocelular carcinoma, as it could predict the response to immune therapy. The manuscript is well-written and the data is clearly presented. I congratulate the authors for this brillant study. 

There is only one minor issue I would like to review. In line 453, page 15, the authors stated that the application of personalized

chemotherapy based on the MAN score would lead to curative effects. I would suggest to soften this sentence, saying that it might lead to better therapeutical responses. 

Author Response

General Comment: Chen et al performed an interesting study in which they developed  the mitochondria-associated endosplaamic reticulum membrane (MAM) score. This score, associated with newly constructed tumor-microenvironment score, was shown to determine the prognosis of patients with hepatocelular carcinoma, as it could predict the response to immune therapy. The manuscript is well-written and the data is clearly presented. I congratulate the authors for this brillant study.

Response: We appreciate the reviewer’s recognition of our work. In response to the issues, we have made the following revisions.

 Comment 1: There is only one minor issue I would like to review. In line 453, page 15, the authors stated that the application of personalized chemotherapy based on the MAN score would lead to curative effects. I would suggest to soften this sentence, saying that it might lead to better therapeutical responses.

Response: Thanks for your suggestion, we have revised this sentence (Revised in Results, line 436-437).

Reviewer 4 Report

The manuscript entitled "Mitochondria-associated endoplasmic reticulum membrane (MAM) is a promising signature to predict prognosis and therapies for hepatocellular carcinoma (HCC)" done by Chen et.al. focused on identifying a new scoring pattern for the prognosis of Hepatocellular carcinomas which can also help in the prediction of the response towards the immune therapy. The basis on which the authors scored is known as a Mitochondrial-associated endoplasmic reticulum membrane (MAM) which signifies the energy metabolic pathways of the cells. The author tried using MAM in order to overcome the currently used method of prognosis which is very general towards all patients but has different outcomes. Therefore the authors suggest a prognosis done on the MAM score. 

The authors have generated these scores based on the MAM-related gene expression which is available in numerous public datasets. The authors validated the MAM score in the TCGA and ICGC data sets. The authors observed that MAM score was higher in the malignant cells using the single-cell data. But a higher MAM score also suggested a better chemotherapy response by the patient. The authors further used Tumor microenvironment scoring to divide the MAM scores. The authors were able to show that a high MAM score/low TME score in the patients has a poor prognosis and higher genomic mutations whereas the prognosis is better if the MAM score is low or the TME is high. Such showed that such patients tend to have a better response to the immune-therapy. 

The authors had a clear hypothesis and the approach used for testing the hypothesis is streamlined by a set experimental design based on the available data sets. The quality of data presented and the statistical test are done appropriately to the best of my knowledge. The materials and methods section is complete and gives all the necessary information. Finally, the discussion is well written and discusses the possible mechanism, and proposes further studies. However, there are some concerns which are as follows.

The authors need to briefly explain more about the MAM gene family or MAM in general in the introduction section so that the readers know about it. 

The authors have observed one of the MAM-related genes namely ACAT1 whose expression is low against a higher MAM score. However, the article PMID”27867011 showed lower levels of ACAT1 are critical and should be targeted for anticancer activity. But the authors have shown that the higher the MAM score the lower the expression of ACAT1 and the lower will be the survivability which is conflicting with the role of ACAT1 in cancer. The authors should briefly explain the mechanism by which ACAT1 can be a good cancer target. They should consider citing their work and also how their results of higher MAM scores and lower ACAT1 help in the better prognosis of cancer patients. 

Overall the work done by Chen et.al. is commendable and adds to the necessary information.

Author Response

lower the expression of ACAT1 and the lower will be the survivability which is conflicting with the role of ACAT1 in cancer. The authors should briefly explain the mechanism by which ACAT1 can be a good cancer target. They should consider citing their work and also how their results of higher MAM scores and lower ACAT1 help in the better prognosis of cancer patients.

Response: Thank you for your suggestion. We did observe that ACAT1 is indeed highly expressed in lung cancer, head and neck cancer, leukemia, and prostate cancer in this article (PMID:27867011). However, we found that expression of ACAT1 is low expressed in HCC datasets (10/11), and high expression of ACAT1 is associated with longer prognosis. We believe that this may be due to the high heterogeneity of tumors, and perhaps the molecule specificity of ACAT1 is lowly expressed in HCC. We greatly appreciate your point, which is needed for further investigation. In addition, we have added this section in the article (Revised in Discussion, line 455-459).